# Mind the Interference: Towards Robust Continual Learning Across Modalities

## Abstract

Continual learning aims to learn knowledge of tasks observed in sequential time steps while mitigating the forgetting of previously learned knowledge. Existing methods were designed to learn a single modality (e.g., image) over time, which limits their applicability in scenarios involving multiple modalities. In this work, we propose a novel continual learning framework that accommodates multiple modalities (*image, video, audio, depth, and text*). We train a model to align various modalities with text, leveraging its rich semantic information. However, this increases the risk of forgetting previously learned knowledge, exacerbated by the differing input traits across tasks. To alleviate the overwriting of previous knowledge of modalities, we propose a framework that consolidates intra-modal knowledge while incorporating relevant inter-modal information. This is achieved by self-regulating shifts in learned representations to gradually integrating novel knowledge into the information retained across modalities. Simultaneously, it mitigates inter-modal interference by selectively integrating knowledge from previously encountered modalities based on their mutual relevance. Furthermore, we introduce a strategy to re-align modality embeddings, effectively addressing biased alignment between modalities. We evaluate the proposed method in a wide range of continual learning scenarios using multiple datasets with different modalities. Extensive experiments demonstrate that ours outperforms existing methods in the scenarios, regardless of whether the identity of the modality is given.

## 1 Introduction

Recently, a vision-language model, pre-trained with a large number of image-text pairs, has drawn significant attention Radford et al. (2021) due to its strong generalization performance Lin et al. (2023); Rasheed et al. (2023). Leveraging the general knowledge of the pre-trained model, several algorithms have achieved promising results in image classification Khattak et al. (2023), semantic segmentation Lüddecke & Ecker (2022), and video action recognition Rasheed et al. (2023), to name a few. However, when deep learning models are exposed to new data, they tend to overwrite previously learned knowledge, leading to forgetting earlier information. This issue is known as catastrophic forgetting McCloskey & Cohen (1989); Kirkpatrick et al. (2017).

To alleviate this phenomenon, continual learning methods strive to preserve existing knowledge while accommodating new information Kirkpatrick et al. (2017); Aljundi et al. (2018); Rebuffi et al. (2017); Smith et al. (2023a); Hou et al. (2018); Li & Hoiem (2016). To this end, they apply penalties to minimize changes in key parameters that contribute to performance on previous tasks Kirkpatrick et al. (2017); Aljundi et al. (2018) or distill the knowledge of the old model into newer ones Hou et al. (2018); Li & Hoiem (2016). Furthermore, to facilitate recall of previous knowledge, some methods rely on having access to data from old tasks Ostapenko et al. (2019); Rebuffi et al. (2017); Douillard et al. (2022); Liu et al. (2021). The collected data is then used jointly with new data to train the model. However, storing raw data can lead to privacy concerns in many applications Smith et al. (2023b). Moreover, temporal data (e.g., video and audio) demand more memory resources than non-temporal data Jin et al. (2023), posing additional challenges for continual learning.

Recently, continual learning methods that adapt a pre-trained model (e.g., ViT Dosovitskiy et al. (2020) or CLIP Radford et al. (2021)) have been proposed Wang et al. (2022b; 2023); He et al. (2025). These methods are designed to select prompts from a shared pool for each task. The selected prompts are inserted into a pre-trained model and updated to acquire new knowledge. Despite

Figure 1: Comparison between (a) unimodal and (b–d) multimodal continual learning sequences addressed in this paper. Unlike the unimodal setting (a), the multimodal setting can introduce diverse modality combinations (b-d), which significantly complicates the retention of previously acquired knowledge while integrating information from tasks across diverse modalities.

outperforming other methods by leveraging the rich knowledge of the pre-trained model, these approaches validate on a single modality and do not account for data from new modalities. This limitation restricts their applicability in dynamic environments where data from different modalities appear, such as in autonomous driving Prakash et al. (2021) and robot planning Huang et al. (2025). More importantly, these approaches are susceptible to forgetting knowledge learned previously when learning multiple modalities because modality gaps between tasks cause prompts trained on different input types to interfere with each other.

In this paper, we propose a novel approach, **COMM** (**CO**ntinual Learning for **M**ultiple **M**odalities), for continuously learning tasks of different modalities, using a pre-trained vision-language model. Specifically, we take text-bound data as input, leveraging the rich semantic information provided by text Zhu et al. (2023). We process the text data through a language encoder and the other modalities through a modality encoder, enabling the integration of additional modalities without introducing many parameters. To leverage rich pre-trained knowledge, we introduce prompts to accommodate new knowledge without updating the backbone parameters. These prompts are accumulated over time for each modality through self-regularization, which preserves representational consistency across time and supports stable integration of intra-modality knowledge. To reduce interference across modalities, we promote modality-aware adaptation by aggregating prompts from previously encountered modalities, suppressing the influence of unrelated ones. Additionally, to prevent biased alignment of recently learned text Wu et al. (2019), we re-align earlier modality representations to restore semantic consistency with their textual counterparts. Figure 1 illustrates the proposed *multimodal continual learning* compared to existing unimodal continual learning.

We demonstrate our method by comparing it with existing continual learning methods across tasks involving multiple modalities using classification benchmark datasets: ImageNet-100 Deng et al. (2009), UCF-101 Soomro et al. (2012), SUN-RGBD Song et al. (2015), and ESC-50 Piczak (2015). Experimental results show that the proposed method outperforms its competitors across various class-incremental learning scenarios while introducing negligible memory overhead from the prompts. The contributions of our work are four-fold: (**i**) We propose a novel multimodal continual learning method alleviates the challenge of intra- and inter-modality interference that arises from learning the distinct traits of each modality in sequential data. (**ii**) We present a knowledge aggregation approach that effectively learns a new task by mitigating interference between modalities while preserving existing knowledge with negligible memory overhead. (**iii**) Our re-alignment strategy restores distorted semantic consistency between previously learned modalities and text during adaptation to new tasks. (**iv**) Extensive experiments show that the proposed method achieves outstanding performance compared to existing continual learning methods by effectively preventing detrimental knowledge mixing across modalities.

## 2 RELATED WORK

**Multimodal learning** aims to construct unified representations from diverse data types, with significant progress driven by pre-trained models such as CLIP Radford et al. (2021). These models are commonly adapted to downstream tasks, including image classification Khattak et al. (2023); Zhou et al. (2022); Wortsman et al. (2022) and video-text matching Lin et al. (2022); Rasheed et al. (2023), via fine-tuning pre-trained weights or inserting learnable prompts. Recent studies Girdhar et al. (2023); Zhu et al. (2023); Zhou et al. (2025) have extended this framework to encompass three

or more modalities, including image and text. While these approaches train modalities jointly, sequential training poses a greater challenge, as it must preserve representations of previously learned modalities while integrating new ones. Moreover, existing methods often rely on modality-specific networks Girdhar et al. (2023); Zhu et al. (2023), limiting scalability as the number of modalities grows. In this work, we present a global encoder for non-text modalities that aligns prompts across modalities, mitigating interference and enhancing representational consensus.

**Continual learning** aims to maintain previously learned knowledge while simultaneously acquiring new information. Existing continual learning methods have primarily focused on a single modality (e.g., image). Early works proposed to identify contributing parameters for previous tasks Kirkpatrick et al. (2017); Friedemann et al. (2017); Aljundi et al. (2018). These methods train the model by penalizing updates to key parameters, which alleviates forgetting of previously learned knowledge. However, they can cause significant loss of knowledge as the number of tasks with different semantic domains increases De Lange et al. (2021). Several studies store Douillard et al. (2022); Liu et al. (2021); Park et al. (2021) or generate Ostapenko et al. (2019); Shin et al. (2017); Wang et al. (2019) some data from old tasks and train the network with a joint set of old and new data. However, privacy or memory budget issues can arise from storing old data Villa et al. (2022). Importantly, the unimodal focus of these approaches limits their applicability in real-world settings where data arrives as a continuous stream from multiple modalities.

**Continual learning with a vision-language model** Radford et al. (2021) has recently been explored, with methods adapting to new unimodal data by matching it with corresponding text descriptions Wortsman et al. (2022); Villa et al. (2023); Wang et al. (2023); Wu et al. (2025). Some methods Zheng et al. (2023); Wortsman et al. (2022) directly update the parameters of a model while regularizing to retain its learned information. The others Villa et al. (2023); Wang et al. (2023); Qiao et al. (2024) learn new tasks by combining trainable prompts with the input embeddings in each layer of a pre-trained model, selecting prompts from a predefined pool. However, these methods often overwrite previously learned knowledge by updating model parameters or prompts, exacerbating forgetting due to interference from different modalities across tasks. In contrast, we aggregate prompts from modalities relevant to the input, ensuring that knowledge learned from other modalities remains unchanged, alleviating interference between modalities.

## 3 METHODOLOGY

### 3.1 PROBLEM DEFINITION

We present a new problem of learning multiple modalities over a sequence of tasks, where at each time step $t$, a subset of modalities, $\mathcal{M}^t \subseteq \mathcal{M} = \{$image, video, depth, audio, ...$\}$, is presented. Specifically, at time step $t$, we learn knowledge of each modality $m \in \mathcal{M}^t$ by aligning it with the corresponding class name (represented by text) in a joint embedding space. The goal of multimodal continual learning is to maintain previously learned knowledge (modality-text alignments) while integrating new knowledge from the current task. At time step $t$, the task[1] is defined by a set of modality-text pairs $T^t = \{D_m^t, C_{\text{text}}^t\}_{m \in \mathcal{M}^t}$. For each modality $m$, $D_m^t$ comprises data points and their labels $\{x_{m,i}^t, y_i^t\}_{i=1}^{n_m^t}$, and $C_{\text{text}}^t$ provides the corresponding textual class names $\{x_{\text{text},i}^t\}_{i=1}^{n_m^t}$.

**Modality-text alignment.** To extract embeddings of the modality data $x_{m,i}^t$ and the text data $x_{\text{text},i}^t$, we use the modality[2] and language encoders, $\boldsymbol{V}(\cdot)$ and $\boldsymbol{L}(\cdot)$, in a pre-trained model Radford et al. (2021), respectively. This approach helps align embeddings from non-text modalities with those from text Girdhar et al. (2023); Zhu et al. (2023). To mitigate the increase in memory cost as the number of modalities grows, non-text modalities share a common modality encoder. We obtain the modality feature $\tilde{v}_{m,i}^t = \boldsymbol{V}(x_{m,i}^t)$ and the text embedding $l_{\text{text},i}^t = \boldsymbol{L}(x_{\text{text},i}^t)$, respectively. $\tilde{v}_{m,i}^t$ is further projected to a common modal-text embedding space as $v_{m,i}^t = \alpha_m^t(\tilde{v}_{m,i}^t)$, by the projection head $\alpha_m^t(\cdot)$. To learn the modality data, we maximize the predictive probability defined as

$$p(y_i^t | x_{m,i}^t) = \frac{\exp(\text{sim}(v_{m,i}^t, l_{\text{text},i}^t)/\tau)}{\sum_j \exp(\text{sim}(v_{m,i}^t, l_{\text{text},j}^t)/\tau)}, \quad (1)$$

where $\text{sim}(\cdot)$ and $\tau$ represent the cosine similarity and the temperature parameter, respectively.

---

[1] We assume that a task appears at every time step.

[2] We implement the modality encoder using the vision encoder in Radford et al. (2021).

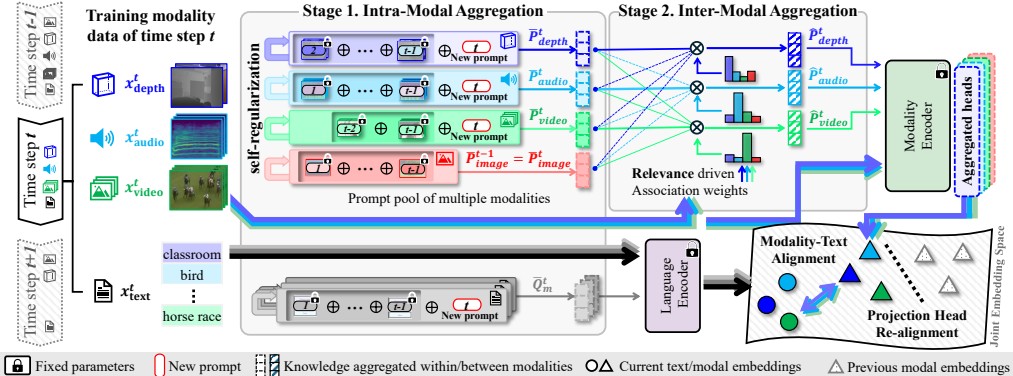

Figure 2: An illustration of the proposed method for learning multiple modalities at time step $t$. The model receives modality data paired with the corresponding text. First, new prompts are introduced and aggregated with previously accumulated ones via self-regularization to preserve intra-modal knowledge. Then, intra-modal prompts are integrated into final prompts using modality-aware relevance weights, which suppress cross-modal interference. The resulting prompts are passed into the modality encoder to obtain aligned features. To prevent biased projections toward recently learned texts, projectors are re-aligned to restore semantic consistency with previously learned alignments.

## 3.2 FRAMEWORK

The proposed method learns multimodal tasks based on a pre-trained model. However, directly updating the model on different modalities can exacerbate forgetting due to interference caused by modality gaps; representational discrepancies and conflicting updates from informational imbalance Peng et al. (2022). To resolve it, we take a strategy of introducing prompts to learn new tasks without updating the model itself Khattak et al. (2023). Figure 2 illustrates our continual learning method for multiple modalities.

Specifically, the set of modalities of the task at time $t$ is $\mathcal{M}^t$, where each modality consists of text-paired data. To learn new tasks, we introduce and update a pair of learnable prompts, $P_m^t$ and $Q_m^t$, of the modality and language encoders by incorporating them with the input, respectively. Unlike existing prompt-based continual learning Qiao et al. (2024); Wang et al. (2023), which employs a prompt pool for unimodal tasks, multimodal continual learning requires a shared pool capable of accommodating tasks with diverse modalities. This introduces a challenge in selecting appropriate prompts.

To effectively utilize prompts for each modality and minimize interference with previously learned representations, we propose aggregating prompts intra and inter modalities in the proposed pipeline. In the first stage, to alleviate significant changes in old knowledge across time within a modality, we combine the prompts learned so far with newly introduced ones by a self-regulating strategy. The update of prompts across modalities can lead to knowledge degradation due to input discrepancies; thus, the second stage mitigates forgetting by aligning knowledge through modality-aware associations. We obtain modality and text embeddings using prompts aggregated intra and inter modalities. The embeddings are aligned by updating $P_m^t$ and $Q_m^t$ using the cross-entropy loss, $\mathcal{L}_{\text{task}} = -\sum_i y_i^t \log p(y_i^t | x_{m,i}^t)$.

## 3.3 CONTINUAL LEARNING FOR MULTIPLE MODALITIES

**Intra-modal aggregation via self-regularization.** To maintain earlier representations and ensure effective prompt utilization within each modality, we aggregate prompts over time and self-regulate them. Specifically, prompts $P_m^t$ and $Q_m^t$ are zero-initialized to gradually acquire knowledge of the task. These prompts are then integrated with those from the previous tasks as $\bar{P}_m^t = \oplus(\bar{P}_m^{t-1}, P_m^t)$ and $\bar{Q}_m^t = \oplus(\bar{Q}_m^{t-1}, Q_m^t)$, where $\bar{P}_m^1 = P_m^1$ and $\bar{Q}_m^1 = Q_m^1$. $\oplus$ denotes element-wise summation, and prompts absent in earlier time steps are excluded from this operation. The aggregated prompts are used to extract modality-text embeddings, $V_{i,\bar{P}_m^{t'}}^t \triangleq \boldsymbol{V}(x_{m,i}^t; \bar{P}_m^{t'})$ and $L_{i,\bar{Q}_m^{t'}}^t \triangleq \boldsymbol{L}(x_{\text{text},i}^t; \bar{Q}_m^{t'})$, which are aligned by minimizing the loss, $\mathcal{L}_{\text{task}}$. However, optimizing solely for the task loss may

fail to adequately preserve prior knowledge, leading to significant shifts in the accumulated representations.

To address this, we introduce a self-regularization loss, $\mathcal{L}_{\text{self}}$, applied to both the prompts and the projection heads, enforcing consistency by minimizing the representation change between outputs from the current and previously aggregated components. Projection heads are similarly aggregated over time: $\bar{\alpha}_m^t = \oplus(\bar{\alpha}_m^{t-1}, \alpha_m^t)$. The complete self-regularization loss is then defined as

$$\mathcal{L}_{\text{self}} = \sum_i \underbrace{\|V_{i,\bar{P}_m^t}^t - V_{i,\bar{P}_m^{t-1}}^t\|_2}_{\text{modality feature consistency}} + \underbrace{\|L_{i,\bar{Q}_m^t}^t - L_{i,\bar{Q}_m^{t-1}}^t\|_2}_{\text{text feature consistency}} + \underbrace{\|\bar{\alpha}^t(V_{i,\bar{P}_m^t}^t) - \bar{\alpha}^{t-1}(V_{i,\bar{P}_m^t}^t)\|_2}_{\text{projected feature consistency}}, \quad (2)$$

where the three terms compute the $\ell_2$ distance between the modality, text, and projected features, respectively, obtained using the current components and their counterparts from the previous ones. Incorporating $\mathcal{L}_{\text{self}}$ into $\mathcal{L}_{\text{task}}$ encourages stable knowledge integration into the parameters $\bar{P}_m^t, \bar{Q}_m^t$, and $\bar{\alpha}_m^t$. This strategy enables gradual adaptation of prompts and projection heads while it preserves prior knowledge and prevents interference from incorrect prompt selection.

**Inter-modal integration guided by relevance.** Prompts trained across modalities are more vulnerable to interference when applied in unrelated contexts than those trained for a single modality. This necessitates the identification of relevant modalities from the pool observed thus far, denoted as $\mathcal{M}^{1:t}$. To this end, we introduce a function $\mathcal{R}^t(\cdot) \colon \mathbb{R}^{d_i} \to \mathbb{R}^{|\mathcal{M}^{1:t}|}$, to identify relevant modalities by examining the features extracted from the modality encoder, $\boldsymbol{V}(\cdot)$. The function assigns probabilities to the prompts, representing their relevance and facilitating the integration of prompts from the modalities $\{\bar{P}_{m'}^t\}_{m' \in \mathcal{M}^{1:t}}$. Due to the inaccessibility of old data, we collect features for each modality, which are sampled from a normal distribution defined by the mean and covariance of the previously observed features Zhang et al. (2023). The function $\mathcal{R}^t(\cdot)$ is trained by minimizing

$$\mathcal{L}_{\text{cross}} = \sum_{m' \in \mathcal{M}^{1:t}} \sum_{i=1}^{\bar{n}_{m'}} -\log \mathcal{R}^t(m'|\bar{u}_{m',i}), \quad (3)$$

where $\bar{u}_{m',i}$ is a collected sample and $\bar{n}_{m'}$ denotes the number of samples.

By minimizing this loss, $\mathcal{R}^t$ yields relevance of each modality $m' \in \mathcal{M}^{1:t}$ in relation to $m$. The prompts are then aggregated across modalities using the relevance $\mathcal{R}^t$ as weights:

$$\hat{P}_m^t = \sum_{m' \in \mathcal{M}^{1:t}} \mathcal{R}^t(m'|\boldsymbol{V}(x_{m,i}^t)) \cdot \bar{P}_{m'}^t, \quad (4)$$

This composed prompt $\hat{P}_m^t$ selectively emphasizes prompts associated with modality contexts relevant to the current input, while attenuating the influence of unrelated ones. Note that the text prompts do not require such aggregation because text data is input solely into the language encoder. By aligning the modality and text embeddings extracted using the prompts, $\hat{P}_m^t$ and $\bar{Q}_m^t$, we can avoid interference between modalities. To summarize, we update the set of parameters $\{P_m^t, Q_m^t, \alpha_m^t\}$ by minimizing the aforementioned loss functions.

**Modality-text re-alignment.** Learning new knowledge without accessing data from old tasks can cause the embeddings from the modality encoder to be biased toward the text embeddings of the latest task Yu et al. (2020), which distorts the previously modality-text alignments. This degrades the semantic grounding of accumulated prompts, as they are composed based on earlier alignments, making the modality-text inconsistency more severe. To address this issue, we introduce an additional phase to re-align the projection head, $\bar{\alpha}_m^t(\cdot)$, so that old and new modality embeddings are distinguished. Specifically, for each class of modality $m$, we sample features that mimic old ones using the class-wise mean and covariance of previously stored embeddings, which are extracted via the modality encoder with the prompt, $\hat{P}_m^i$, where $i < t$. The aggregated projection head, $\bar{\alpha}_m^t$, is further fine-tuned to align the modality embedding derived from the sampled features with the text embedding by minimizing $\mathcal{L}_{\text{task}}$.

| Method | Image | | | Video | | | Depth | | | Audio | | | Overall | | |
|---|---|---|---|---|---|---|---|---|---|---|---|---|---|---|---|
| | AIA (↑) | FAA (↑) | F (↓) | AIA (↑) | FAA (↑) | F (↓) | AIA (↑) | FAA (↑) | F (↓) | AIA (↑) | FAA (↑) | F (↓) | AIA (↑) | FAA (↑) | F (↓) |
| **Modality-Specific Class-incremental Learning** | | | | | | | | | | | | | | | |
| FT | 45.66 | 23.46 | 23.37 | 45.25 | 13.16 | 33.87 | 26.21 | 7.58 | 19.72 | 28.96 | 4.20 | 26.31 | 36.52 | 12.16 | 25.82 |
| EWC Kirkpatrick et al. (2017) | 47.16 | 23.42 | 24.98 | 48.59 | 20.76 | 29.37 | 26.38 | 13.91 | 13.20 | 26.68 | 5.50 | 20.50 | 37.20 | 15.90 | 22.51 |
| LwF Li & Hoiem (2016) | 53.13 | 24.78 | 29.84 | 48.59 | 23.26 | 26.74 | 30.64 | 11.43 | 20.34 | 31.89 | 9.75 | 23.53 | 41.06 | 17.31 | 25.11 |
| WISE-FT Wortsman et al. (2022) | 46.73 | 21.96 | 26.08 | 46.99 | 21.27 | 27.14 | 26.93 | 5.27 | 22.93 | 22.69 | 8.25 | 15.34 | 35.83 | 14.18 | 22.87 |
| L2P Wang et al. (2022b) | 76.96 | 68.66 | 8.74 | 78.49 | 70.71 | 8.21 | 45.06 | 24.75 | 24.66 | 44.28 | 17.00 | 28.98 | 61.20 | 45.28 | 17.65 |
| S-liPrompts Wang et al. (2022a) | 75.61 | 65.25 | 10.91 | 82.85 | 69.04 | 14.57 | 47.98 | 26.08 | 30.07 | 42.68 | 22.25 | 21.70 | 62.28 | 45.66 | 19.31 |
| AttriCLIP Wang et al. (2023) | 77.39 | 65.70 | 12.30 | 77.40 | 70.51 | 7.28 | 51.21 | 35.01 | 17.16 | 43.44 | 20.50 | 24.37 | 62.36 | 47.93 | 15.27 |
| PGP Qiao et al. (2024) | 78.88 | 68.58 | 10.84 | 83.14 | 71.46 | 12.33 | 48.67 | 25.58 | 24.45 | 47.69 | 26.25 | 22.78 | 64.60 | 47.98 | 17.60 |
| COMM (Ours) | **86.07** | **78.96** | **7.11** | **91.03** | **83.82** | **7.21** | **62.90** | **44.03** | 18.88 | **61.86** | **43.25** | 18.60 | **75.47** | **62.51** | 13.71 |
| **Modality-Agnostic Class-incremental Learning** | | | | | | | | | | | | | | | |
| FT | 33.39 | 1.46 | 33.61 | 27.90 | 0.14 | 29.30 | 25.70 | 7.59 | 19.18 | 23.65 | 4.25 | 20.61 | 27.66 | 3.36 | 25.68 |
| EWC Kirkpatrick et al. (2017) | 38.23 | 5.60 | 34.34 | 27.46 | 3.63 | 25.15 | 24.04 | 13.91 | 10.73 | 22.77 | 5.50 | 18.34 | 28.13 | 7.15 | 22.14 |
| LwF Li & Hoiem (2016) | 48.74 | 21.14 | 29.05 | 19.82 | 1.17 | 22.43 | 23.01 | 6.74 | 17.22 | 19.01 | 9.00 | 10.70 | 27.65 | 9.51 | 19.85 |
| WISE-FT Wortsman et al. (2022) | 38.43 | 10.34 | 29.57 | 24.89 | 0.36 | 25.88 | 25.94 | 5.27 | 21.67 | 22.46 | 6.25 | 17.22 | 27.93 | 5.55 | 23.59 |
| L2P Wang et al. (2022b) | 79.85 | 69.84 | 10.53 | 71.25 | 65.14 | 6.44 | 39.92 | 26.72 | 13.96 | 33.08 | 21.50 | 12.31 | 56.03 | 45.80 | 19.31 |
| S-liPrompts Wang et al. (2022a) | 77.36 | 66.72 | 11.20 | 79.52 | 73.18 | 6.69 | 40.94 | 30.02 | 11.56 | 33.63 | 23.50 | 10.79 | 57.87 | 48.36 | 10.06 |
| AttriCLIP Wang et al. (2023) | 73.93 | 63.90 | 10.56 | 73.28 | 68.70 | 4.83 | 39.52 | 28.57 | 11.60 | 34.99 | 22.50 | 13.27 | 55.43 | 45.91 | 10.06 |
| PGP Qiao et al. (2024) | 81.29 | 70.92 | 10.85 | 77.55 | 67.18 | 10.94 | 45.61 | 26.25 | 20.49 | 46.15 | 21.50 | 26.19 | 62.37 | 41.58 | 17.12 |
| COMM (Ours) | **83.89** | **76.84** | **5.43** | **90.83** | **83.63** | 7.60 | **62.78** | **43.94** | 19.84 | **61.85** | **43.25** | 19.60 | **74.84** | **61.92** | 13.19 |

Table 1: Results for the first sequence on the modality-specific (top) and modality-agnostic (bottom) scenarios. The AIA, FAA, and F metrics are measured after training all tasks. Overall denotes the average AIA and FAA across all modalities.

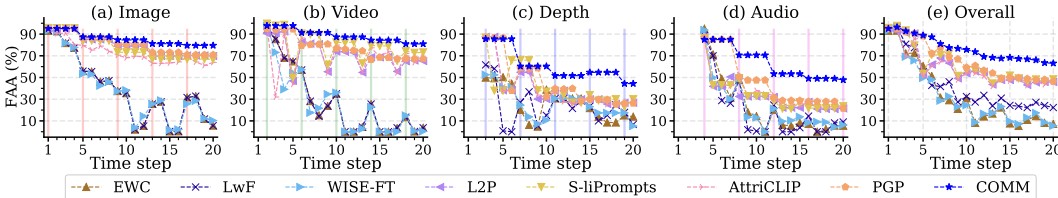

Figure 3: Results of modality-agnostic continual learning for the first sequence, where FAA is measured at each time step. The vertical lines represent the time steps at which the modality is learned.

## 4 EXPERIMENTS

### 4.1 EXPERIMENTAL SETUP

**Scenarios.** We evaluated the proposed method for continual learning of five modalities: image, video, audio, depth, and text, using four datasets: ImageNet-100 Deng et al. (2009), UCF-101 Soomro et al. (2012), ESC-50 Piczak (2015), and SUN-RGBD Song et al. (2015). Each dataset was divided into five non-overlapping subsets of classes. We consider each subset as a task. For a comprehensive evaluation, we designed a benchmark based on three sequences of tasks, as illustrated in Figure 1 (b-d): (i) A random modality is introduced at each time step. (ii) A modality is trained across multiple time steps before shifting to the next modality. (iii) All modalities are available at every time step. The first two sequences comprise 20 time steps (corresponding to five subsets of four modalities), with each task exclusively learning a single modality. In contrast, the third sequence consists of five time steps, during which all modalities are learned simultaneously for every step. For each sequence, we evaluated the methods with and without the provision of modality identity during the evaluation phase, which we refer to as the *modality-specific* and *modality-agnostic* scenarios, respectively. We report average incremental accuracy (AIA) Rebuffi et al. (2017), final average accuracy (FAA) Douillard et al. (2022), and average forgetting (F) Chaudhry et al. (2019).

**Baselines.** We evaluated the proposed method, COMM, with existing continual learning methods: conventional approaches, including sequential fine-tuning (FT), EWC Kirkpatrick et al. (2017), LwF Li & Hoiem (2016), and WISE-FT Wortsman et al. (2022); and prompt-based methods, L2P Wang et al. (2022b), S-liPrompts Wang et al. (2022a), AttriCLIP Wang et al. (2023), and PGP Qiao et al. (2024).

**Implementation details.** For a fair comparison with other continual learning methods, we adopted ViT-B-16 CLIP Radford et al. (2021) as the backbone model for the compared methods. We used the vision and text encoders of CLIP for the non-text and text modalities, respectively, in accordance with established practices Girdhar et al. (2023); Zhu et al. (2023). We used a fully connected layer for $\mathcal{R}^t(\cdot)$. The proposed method was trained using the Adam optimizer Kingma & Ba (2015) with $\beta_1$ and $\beta_2$ of 0.9 and 0.999. Additional information on datasets, preprocessing per modality, baseline setups, and hyperparameters are provided in the appendix.

| Method | Modality-Specific Learning | | | | | | Modality-Agnostic Learning | | | | | |
|---|---|---|---|---|---|---|---|---|---|---|---|---|
| | Image (↑) | Video (↑) | Depth (↑) | Audio (↑) | Overall AIA (↑) | Overall F (↓) | Image (↑) | Video (↑) | Depth (↑) | Audio (↑) | Overall AIA (↑) | Overall F (↓) |
| FT | 26.38 | 27.27 | 20.00 | 26.39 | 25.01 | 24.56 | 26.05 | 22.61 | 16.45 | 25.74 | 22.71 | 22.64 |
| EWC Kirkpatrick et al. (2017) | 27.21 | 28.45 | 18.19 | 30.32 | 26.04 | 24.86 | 26.10 | 24.37 | 16.98 | 29.88 | 24.33 | 26.59 |
| LwF Li & Hoiem (2016) | 26.92 | 28.52 | 18.71 | 27.76 | 25.48 | 24.83 | 20.48 | 23.12 | 18.23 | 24.45 | 21.57 | 21.49 |
| WISE-FT Wortsman et al. (2022) | 27.08 | 27.85 | 16.53 | 29.17 | 25.16 | 24.67 | 23.43 | 24.00 | 16.01 | 28.02 | 22.87 | 24.63 |
| L2P Wang et al. (2022b) | 70.73 | 72.96 | 32.25 | 39.54 | 53.87 | 11.25 | 72.19 | 62.37 | 37.03 | 43.38 | 53.74 | 11.95 |
| S-liPrompts Wang et al. (2022a) | 67.93 | 75.45 | 40.88 | 43.25 | 56.88 | 12.02 | 67.81 | 73.42 | 32.17 | 38.62 | 53.01 | 11.29 |
| AttriCLIP Wang et al. (2023) | 68.62 | 72.69 | 42.30 | 42.24 | 56.46 | 10.43 | 67.89 | 69.91 | 38.26 | 44.05 | 55.03 | 12.62 |
| PGP Qiao et al. (2024) | 72.51 | 76.51 | 37.42 | 43.87 | 57.58 | 10.11 | 72.15 | 66.24 | 37.22 | 42.72 | 54.58 | 11.31 |
| COMM (Ours) | **81.95** | **85.17** | **50.39** | **61.01** | **69.63** | **7.81** | **80.35** | **84.99** | **50.30** | **60.94** | **69.14** | **7.94** |

Table 2: Results for the second sequence. AIA and F are measured after training on all tasks. Overall AIA and Overall F represent the average AIA and average F across all modalities, respectively.

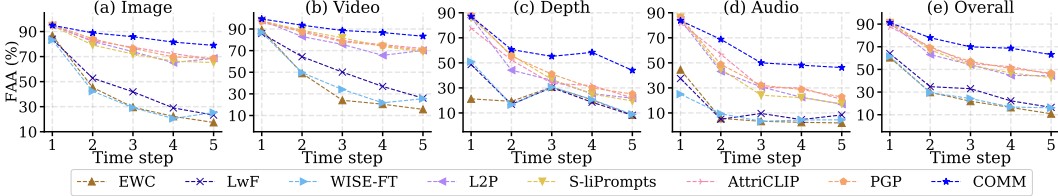

Figure 4: Results (measured in FAA) of multimodal continual learning for the third sequence.

## 4.2 MAIN RESULTS

**Sequence 1.** We evaluated our method and the compared methods on the first sequence for multimodal continual learning. Table 1 summarizes the results of the modality-specific (top) and modality-agnostic (bottom) class-incremental learning scenarios. Overall, we observe that the results of the modality-agnostic scenario show lower performance compared to the modality-specific scenario due to the challenge in identifying the modality. The regularization methods, EWC, LwF, and WISE-FT, perform poorly in both scenarios compared to the prompt-based continual learning methods, L2P, S-liPrompts, AttriCLIP, PGP, and COMM. This indicates that directly updating the network parameters in the regularization methods to accommodate knowledge from multiple modalities is prone to forgetting previously learned tasks due to contaminated mixed knowledge. Even though the existing prompt-based methods outperform other continual learning approaches, they do not perform well for the depth and audio modalities. This is mainly due to the absence of pre-training on non-visual data, which leads to greater forgetting; the proposed method alleviates this by effectively weighting prompts and suppressing cross-modal interference. Notably, these methods exhibit lower AIA in the modality-agnostic scenario due to the challenge of selecting prompts from the shared pool, which hinders learning new tasks. The proposed method, aiming to alleviate sensitivity to prompt selection and biased projections, outperforms the best competitor, S-liPrompts, with an overall FAA gap of 16.85% and 13.56% in the respective scenarios. In terms of the performance gap between the scenarios, the proposed method shows only a marginal performance gap of 0.63% in overall AIA, in contrast to the larger gaps of 4.41% and 6.93% observed with S-liPrompts and AttriCLIP, respectively. Additional results for different class orders can be seen in the appendix.

We also present detailed results of the modality-agnostic scenario (measured in FAA) of the compared methods in Figure 3. The conventional methods that directly update the network parameters exhibit high fluctuations. The performance of each modality slightly increases when tasks with the specific modality are learned (indicated by vertical lines). However, learning tasks from other modalities (indicated by non-vertical lines) results in a significant performance decline due to overwritten knowledge. The prompt-based continual learning methods, L2P, S-liPrompts, AttriCLIP, and PGP, also show unsatisfying performance in video, depth, and audio modalities. In contrast, the proposed method exhibits stable performance across different modalities, surpassing other approaches, particularly in the audio and depth modalities.

**Sequence 2.** To further evaluate the consistency of our approach, we compared COMM with other continual learning methods in another sequence, where each modality is trained over multiple times before shifting to another (in order of image, video, depth, and audio). Table 2 reports the experimental results for the sequence, demonstrating a similar performance trend to that observed in Table 1. However, conventional continual learning methods show a significant performance degradation for modalities learned early. Specifically, the image and video modalities experience about a 20% drop in AIA compared to the first sequence. Similarly, prompt-based methods show per-

| Method | | | First Sequence (Random) | | | | | Second Sequence (Shift) | | | | |
|---|---|---|---|---|---|---|---|---|---|---|---|---|
| Cross | Self | Re-align | Image | Video | Depth | Audio | Overall | Image | Video | Depth | Audio | Overall |
| ✓ | - | - | 62.81 | 80.42 | 46.44 | 49.84 | 59.88 | 59.77 | 70.14 | 23.35 | 43.76 | 49.26 |
| - | ✓ | - | 73.95 | 75.14 | 48.92 | 59.35 | 64.34 | 65.52 | 65.94 | 30.75 | 56.24 | 54.61 |
| - | - | ✓ | 76.10 | 81.50 | 49.32 | 53.65 | 65.14 | 67.81 | 76.88 | 31.92 | 51.11 | 56.93 |
| ✓ | ✓ | - | 71.91 | 73.21 | 48.51 | 59.53 | 63.29 | 64.08 | 65.78 | 41.76 | 54.46 | 56.53 |
| ✓ | - | ✓ | 76.40 | 81.47 | 58.01 | 54.99 | 67.72 | 66.73 | 76.63 | 25.65 | 49.77 | 54.70 |
| - | ✓ | ✓ | 83.05 | 88.62 | 47.55 | 60.59 | 69.95 | 78.17 | 81.51 | 29.92 | 60.83 | 62.61 |
| ✓ | ✓ | ✓ | **83.89** | **90.83** | **62.78** | **61.85** | **74.84** | **80.35** | **84.99** | **50.30** | **60.94** | **69.14** |

Table 3: Results of the ablation study on the usage of the components in the proposed method.

formance declines for previously learned modalities; however, they achieve enhanced performance for the most recently acquired modality, audio. The results indicate that learning different modalities increases interference between modalities, leading to more forgetting in both continual learning methods. COMM outperforms PGP and AttriCLIP in each scenario, demonstrating significant AIA improvements in audio with gaps of 17.14% and 16.89%, respectively. Additionally, it achieves overall AIA improvements of 12.05% (with an overall F decrease of 2.30%) and 14.11% (with an overall F decrease of 4.68%) compared to PGP and AttriCLIP. We also provide additional results on sequences with varying task orders in the appendix.

**Sequence 3.** We report the results for multiple modalities learned simultaneously at every step in a modality-specific scenario. Figure 4 shows the results for the sequence. The results reveal a clear difference between methods using prompts (L2P, S-liPrompts, AttriCLIP, and PGP) and those updating model parameters (EWC, LwF, and WISE-FT) across all time steps. This indicates that directly updating model parameters for multiple modalities can lead to severe conflicts, resulting in poor performance. The proposed approach consistently demonstrates higher performance across all modalities than the competitors due to reduced interference between modalities.

## 4.3 MORE RESULTS

**Ablation study on the components**. We conducted an ablation study of the proposed method, including its components: cross-modality integration (*Cross*), self-regularization (*Self*), and projection head re-alignment (*Re-align*). Note that the method without *Cross* or *Self* requires identifying the modality identity or task to select prompts, respectively. Therefore, we employed a prompt selection mechanism in Wang et al. (2022a). We compared these methods to the first two sequences in the modality-agnostic scenario. Table 3 reports the results of the ablation study with AIA measured after training all tasks. Overall, the proposed method using an individual component results in unsatisfactory performance on both task sequences compared to employing them together. The method using *Cross* identifies relevant modalities but struggles to select accurate task-specific prompts. When comparing the method utilizing all components against that excluding *Cross*, it becomes evident that selecting suitable modality prompts significantly enhances performance, with gaps of 15.23% and 20.38% in the depth modality for the first and second sequences, respectively. The method that employs both *Self* and *Re-align* exhibits a strong synergistic effect because *Self* preserves previous features by regularizing prompts, and these preserved features closely resemble those used by *Re-align*. We provide visualizations demonstrating the maintainability of *Self* in the embedding space in the appendix.

**Study of scalability.** In this study, we evaluated the scalability of the proposed method by comparing the parameter growth of COMM with other prompt-based methods in Figure 5. Overall, prompt-based methods require fewer parameters than updating the entire model Kirkpatrick et al.

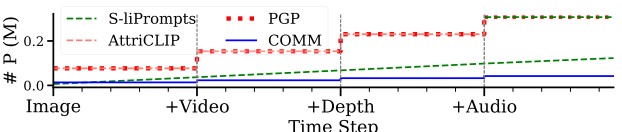

Figure 5: The number of parameters with respect to the modality.

(2017); Li & Hoiem (2016), which involves approximately 124 million parameters. Specifically, L2P and AttriCLIP leverage a prompt pool for each modality, resulting in a step-wise increase in parameters as new modalities are introduced. S-liPrompts uses task-specific prompts, leading to a monotonic parameter increase. The proposed method, COMM, consolidates prompts inter- and intra-modalities, enhancing scalability when integrating additional modalities. Additional results for

| Method | Prompt Insertion | Intra-modal | Inter-modal | Image | Video | Depth | Audio | Overall |
|---|---|---|---|---|---|---|---|---|
| COMM-VL (Ours) | Both encoders | Self-Regulation | Learned relevance | **86.07** | **91.03** | **62.90** | **61.86** | **75.47** |
| COMM-L | Language encoder | Self-Regulation | Learned relevance | 85.30 | 83.89 | 61.56 | 60.43 | 72.80 |
| COMM-V | Modality encoder | Self-Regulation | Learned relevance | 85.76 | 90.50 | 59.22 | 61.35 | 74.21 |
| COMM-Sum | Both encoders | Sum | Learned relevance | 78.00 | 77.70 | 51.10 | 52.90 | 65.00 |
| COMM-Avg | Both encoders | Avg | Learned relevance | 76.90 | 76.60 | 51.70 | 54.40 | 64.90 |
| COMM-NP | Both encoders | Self-Regulation | Mahalanobis | 85.10 | 84.70 | 57.80 | 59.80 | 71.80 |

Table 4: Ablation study on prompt configuration. We compare variations in (a) insertion location, (b) intra-modality prompt accumulation strategy, and (c) inter-modal prompt integration method.

the different numbers of learnable parameters and for unimodal continual learning are included in the appendix.

**Study on prompt configuration.** We conducted an additional study on the design choices of the proposed prompting mechanism. Specifically, we analyze how prompts should be configured and composed to retain previously learned knowledge and to support new modalities without interference. This study investigates (1) the location of prompt insertion (language encoder vs. modality encoder), (2) the strategy for accumulating prompts within each modality over time, and (3) the mechanism for composing prompts across modalities based on relevance. The results of this study are summarized in Table 4.

*(1) Prompt Insertion Location:* We evaluate the impact of prompt insertion by comparing three variants: inserting prompts only into the language encoder (COMM-L), only into the modality encoder (COMM-V), or into both (COMM-VL), with the last corresponding to the proposed method. The results show that COMM-VL achieves the highest performance, indicating that using prompts in both encoders is essential for learning modality-specific features while maintaining alignment with semantic labels.

*(2) Intra-Modality Accumulation:* To mitigate forgetting within each modality, we introduce a self-regularized accumulation strategy that incrementally incorporates newly learned prompts while preserving the semantic continuity of previously acquired ones. This approach is compared against two simplified alternatives: element-wise summation (COMM-Sum) and averaging (COMM-Avg) of past prompts. Results indicate that these naïve methods substantially degrade performance, which confirms that uncontrolled accumulation disrupts discriminative prompt representations. The self-regularization strategy is thus essential for ensuring consistency across learning steps during intra-modal knowledge integration.

*(3) Inter-Modality Composition:* To assess the effectiveness of the proposed relevance-guided prompt composition across modalities, we compare it with a non-parametric baseline (COMM-NP) that estimates modality similarity using Mahalanobis distance computed over Gaussian-sampled features. While both methods utilize the same statistical feature summary, the learned relevance scores in our approach yield consistently superior results. This indicates that learning relevance scores enables more precise and robust identification of semantically related modalities, resulting in more stable and effective cross-modal prompt integration.

## 5 CONCLUSION

In this work, we have introduced the novel multimodal continual learning framework that mitigates interference from the modality gap across tasks by consolidating intra- and inter-modality knowledge. Learning a sequence of tasks with different modalities complicates the retention of prior knowledge as accumulating mixed information over time interferes with the learning of individual modalities. To address this challenge, we have proposed a strategy that aggregates knowledge in each modality over time through self-regularization and integrates it across relevant modalities. This approach selectively emphasizes contributions from relevant modalities while minimizing the influence of unrelated ones, gradually assimilating new information and preserving existing knowledge. Extensive experimental results, including modality-specific and modality-agnostic scenarios, demonstrated that ours significantly outperforms other continual learning methods with a notable performance gap.

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
