## A  APPENDIX

### A.1  ADDITIONAL IMPLEMENTATION DETAILS

**Additional details of datasets.** The datasets used for the modalities are ImageNet-100 Deng et al. (2009), UCF-101 Soomro et al. (2012), ESC-50 Piczak (2015), and SUN-RGBD Song et al. (2015), each accompanied by corresponding text data. ImageNet-100 contains 100 classes selected from the original ImageNet dataset Deng et al. (2009), widely used in continual learning for image classification Douillard et al. (2022); Yan et al. (2021). UCF-101 contains 101 action categories for video action recognition Soomro et al. (2012). ESC-50 is a fine-grained collection of environmental audio recordings of 50 classes for sound classification. For the depth modality, we performed on the SUN-RGBD dataset for scene classification.

**Preprocessing multiple modalities.** Images from ImageNet-100 were resized to a spatial dimension of $224 \times 224$ pixels. We applied center cropping for augmentation. For SUN-RGBD, we converted the depth maps into disparity maps following Girdhar et al. (2022) and normalized the disparity map between 0 and 1. For the video dataset, we randomly selected three frames from every video clip of two seconds. The height and width of each video frame were resized to $224 \times 224$ pixels, resulting in dimensions of $6 \times 3 \times 224 \times 224$. We augmented videos with center cropping and horizontal flipping following Villa et al. (2022). For the audio dataset, we selected three audio clips from each audio sample. We processed each audio clip by sampling it at 16kHz, followed by extracting a mel-spectrogram with 224 frequency bins using a 25ms Hamming window every 10ms Girdhar et al. (2023), and resized the spatial dimension to $224 \times 224$, resulting in dimensions of $3 \times 3 \times 224 \times 224$. We applied the same processing and augmentation for the modalities across all comparison methods. The preprocessed non-text modalities were tokenized using the tokenizer of the pre-trained vision encoder Radford et al. (2021), following previous practices Girdhar et al. (2023); Zhu et al. (2023).

**Implementation details.** We inserted prompts up to the fifth layer of each encoder, following Khattak et al. (2023). Additionally, due to the high dimension of the projection layer $\alpha_m^t$, we employed a low-rank decomposition technique Hu et al. (2021) to split it into two smaller matrices with a rank of 20. To train modalities involving temporal information (video and audio), we duplicated the prompts for the temporal dimensions and concatenated them with the spatial features of each temporal dimension Rasheed et al. (2023). We report the average results from three independent runs for all experiments using NVIDIA RTX A5000 GPUs.

The modality features at time $t$, $\boldsymbol{V}(x_{m,i}^t; \hat{P}_m^t)$, were average pooled along the temporal dimension before being fed into the projection head. To train the layer $\mathcal{R}^t$, we computed and stored the mean and covariance of the features for each modality at the beginning of every time step. Specifically, for the $i$-th sample of modality $m$, $x_{m,i}^t$, the feature was calculated as $\boldsymbol{V}(x_{m,i}^t)$. For modality $m$, if the mean and covariance had already been stored from previous tasks, the mean was updated by combining the previous and new means based on their sample sizes, and the covariance was recomputed using the updated mean and new statistics. Finally, we trained the function $\mathcal{R}^t$ using the sampled features to capture the relevance between modalities.

## A.2 Additional Results

**Results on different task orders.** For the experiment conducted in the Table 1 in the main paper, the sequence consists of 20 tasks, where image, video, depth, and audio modalities were learned repetitively. For the sequence in Table 2, modality was trained for five time steps before shifting to next modality in the order of image, video, depth, and audio. To investigate the sensitivity of the compared methods with respect to different task orders in continual learning, we conducted an additional experiments using the reverse order of the tasks used in Tables 1 and 2.

The results are reported in Table 5. The conventional continual learning methods, EWC, LwF, and WISE-FT, are highly sensitive to the order of tasks for both sequences. In contrast, the prompt-based methods, L2P, S-liPrompt, AttriCLIP, PGP, and COMM, show less sensitivity to the order of tasks compared to the conventional methods. The proposed method, COMM, demonstrates robustness to changes in task order, consistently outperforms other continual learning methods. Specifically, the proposed method shows a marginal overall AIA gap of 0.09% between the different task orders in the second sequence. However, other prompt-based continual learning methods, L2P, S-liPrompts, AttriCLIP, and PGP, show larger performance gaps of 3.64%, 2.31%, 5.67%, and 4.18%, respectively.

| Method | Reverse Order of First Sequence (Random) | | | | | |
| | Audio (↑) | Depth (↑) | Video (↑) | Image (↑) | Overall AIA (↑) | Overall F (↓) |
| --- | --- | --- | --- | --- | --- | --- |
| FT | 27.18 | 26.65 | 9.13 | 6.33 | 17.32 | 8.75 |
| EWC Kirkpatrick et al. (2017) | 33.06 | 28.76 | 9.34 | 2.22 | 18.35 | 9.84 |
| LwF Li & Hoiem (2016) | 26.12 | 22.26 | 13.88 | 8.06 | 17.58 | 11.35 |
| WISE-FT Wortsman et al. (2022) | 28.48 | 28.13 | 8.71 | 4.57 | 17.47 | 9.63 |
| L2P Wang et al. (2022b) | 42.43 | 44.01 | 69.35 | 62.80 | 54.65 | 19.29 |
| S-liPrompts Wang et al. (2022a) | 44.03 | 45.94 | 74.03 | 61.14 | 56.29 | 20.01 |
| AttriCLIP Wang et al. (2023) | 44.88 | 42.62 | 68.31 | 58.90 | 53.68 | 20.89 |
| PGP Qiao et al. (2024) | 41.20 | 49.40 | 79.63 | 79.30 | 62.39 | 17.05 |
| COMM (Ours) | **63.08** | **61.22** | **89.14** | **87.03** | **75.12** | 10.47 |
| Method | Reverse Order of Second Sequence (Shift) | | | | | |
| | Audio (↑) | Depth (↑) | Video (↑) | Image (↑) | Overall AIA (↑) | Overall F (↓) |
| FT | 12.24 | 16.89 | 20.13 | 38.98 | 22.06 | 17.93 |
| EWC Kirkpatrick et al. (2017) | 12.44 | 16.56 | 23.78 | 39.42 | 23.05 | 18.25 |
| LwF Li & Hoiem (2016) | 12.90 | 17.45 | 24.67 | 39.91 | 23.73 | 19.16 |
| WISE-FT Wortsman et al. (2022) | 14.16 | 17.42 | 24.67 | 38.57 | 23.71 | 16.81 |
| L2P Wang et al. (2022b) | 24.98 | 31.01 | 66.75 | 77.64 | 50.10 | 6.68 |
| S-liPrompts Wang et al. (2022a) | 24.85 | 31.18 | 71.65 | 75.10 | 50.70 | 12.04 |
| AttriCLIP Wang et al. (2023) | 26.61 | 28.96 | 68.92 | 72.94 | 49.36 | 7.59 |
| PGP Qiao et al. (2024) | 26.08 | 33.56 | 62.13 | 79.82 | 50.40 | 11.37 |
| COMM (Ours) | **53.02** | **53.42** | **84.62** | **85.88** | **69.23** | 4.96 |

Table 5: Results for the reverse task order of the first (left) and second (right) sequences in the modality-agnostic class-incremental learning scenario. The AIA and F is measured after training on all tasks. Overall AIA and Overall F represent the average AIA and average F across all modalities.

**Results on different class orders.** We followed the class order provided by Villa et al. (2022) for the video dataset. Similar to the practices in other continual learning benchmarks Villa et al. (2022); Rebuffi et al. (2017); Wang et al. (2023), we randomly selected the class order for the image Deng et al. (2009), depth Song et al. (2015), and audio Piczak (2015) datasets. Based on the determined class orders, the image and audio datasets were divided into five subsets consisting of 20 and 10 classes, respectively. The video dataset was divided into five subsets, with the first subset comprising 21 classes and the other four subsets containing 20 classes each. Similarly, the depth dataset was divided into five subsets, with the first four subsets comprising eight classes each and the remaining subset containing 12 classes. The class orders for the image, video, depth, and audio data used in the main paper are listed in Table 6.

| Image | Video |
| --- | --- |
| great white shark, drake, electric ray, mud turtle, sea lion, wallaby, conch, American coot, oystercatcher, barn spider, hognose snake, great grey owl, coucal, diamondback, prairie chicken, sea slug, black swan, black and gold garden spider, spiny lobster, leatherback turtle, black grouse, bittern, common newt, white stork, tiger shark, kite (bird of prey), tick, pelican, garden spider, green mamba, horned viper, sea anemone, snail, crayfish, vine snake, sulphur-crested cockatoo, hen, redshank, jellyfish, macaw, chambered nautilus, wolf spider, water ouzel, stingray, Dungeness crab, hammerhead, hermit crab, scorpion, crane, goldfinch, bustard, lorikeet, thunder snake, goldfish, tarantula, cock, agama, goose, ptarmigan, sea snake, flamingo, green snake, toucan, axolotl, chickadee, peacock, night snake, hornbill, tailed frog, common iguana, boa constrictor, red-backed sandpiper, Komodo dragon, spoonbill, black widow, bulbul, whiptail, rock crab, American alligator, flatworm, banded gecko, sidewinder, chiton, spotted salamander, magpie, bee eater, king snake, terrapin, hummingbird, wombat, garter snake, green lizard, tench, albatross, nematode, harvestman, bald eagle, indigo bunting, loggerhead, limpkin. | surfing, salsa spin, playing tabla, pole vault, yo yo, mixing, apply eye makeup, volleyball spiking, long jump, field hockey penalty, rafting, hammering, push ups, handstand walking, blowing candles, golf swing, playing guitar, band marching, brushing teeth, playing flute, front crawl, archery, bench press, punch, horse riding, cutting in kitchen, billiards, cricket bowling, diving, haircut, boxing speed bag, horse race, still rings, high jump, biking, ice dancing, skijet, writing on board, hula hoop, throw discus, parallel bars, playing daf, wall pushups, swing, hammer throw, balance beam, typing, breast stroke, tennis swing, handstand pushups, sky diving, knitting, rope climbing, skiing, basketball dunk, baby crawling, rowing, nunchucks, table tennis shot, trampoline jumping, bowling, basketball, drumming, skate boarding, lunges, baseball pitch, kayaking, cliff diving, frisbee catch, rock climbing indoor, soccer juggling, shot put, playing piano, javelin throw, military parade, boxing punching bag, sumo wrestling, soccer penalty, pommel horse, walking with dog, clean and jerk, apply lipstick, playing violin, jumping jack, fencing, pull ups, juggling balls, body weight squats, pizza tossing, head massage, playing dhol, shaving beard, mopping floor, floor gymnastics, blow dry hair, jump rope, playing cello, playing sitar, tai chi, uneven bars, cricket shot. |
| **Depth** | **Audio** |
| dancing room, computer room, coffee room, lobby, office dining, idk, storage room, bedroom, lecture theatre, library, corridor, reception room, playroom, classroom, cafeteria, gym, study space, indoor balcony, office, basement, living room, rest space, dinette, dining room, lab, furniture store, hotel room, printer room, bathroom, stairs, kitchen, reception, study, dining area, bookstore, exhibition, discussion area, home, mail room, laundromat, home office, recreation room, office kitchen, conference room. | airplane, wind, clock tick, toilet flush, sneezing, rooster, sheep, pig, door wood creaks, hand saw, washing machine, crying baby, train, clock alarm, frog, rain, siren, crickets, snoring, helicopter, keyboard typing, cow, laughing, church bells, coughing, can opening, water drops, vacuum cleaner, insects, cat, door wood knock, thunderstorm, chainsaw, crackling fire, car horn, brushing teeth, footsteps, clapping, pouring water, mouse click, fireworks, breathing, sea waves, glass breaking, hen, chirping birds, crow, dog, drinking sipping, engine. |

Table 6: Class orders (represented by their names) for each modality dataset.

To validate the robustness of the proposed method under different class orders, we conducted additional experiments using three randomly shuffled class orders in the modality-specific scenario for the first sequence. Since regularization-based continual learning approaches exhibit unsatisfactory performance, we omit them from the comparison. Table 7 shows AIA, FAA, and Forgetting (F) results after training on all tasks. COMM consistently outperforms all other methods across all modalities in terms of AIA and FAA while exhibiting the lowest forgetting. Specifically, COMM outperforms the runner-up method, PGP, with performance margins of 12.2%, 15.5%, and 4.4% in

| Method | | L2P | S-liPrompts | AttriCLIP | PGP | COMM |
|---|---|---|---|---|---|---|
| Image | AIA (↑) | 79.3 ± 2.1 | 76.4 ± 0.6 | 77.9 ± 2.4 | 81.0 ± 1.8 | **85.9 ± 0.3** |
| | FAA (↑) | 70.9 ± 2.4 | 67.7 ± 1.8 | 66.6 ± 3.0 | 71.4 ± 2.6 | **79.6 ± 0.4** |
| | F (↓) | 8.8 ± 0.5 | 9.1 ± 1.4 | 11.9 ± 0.7 | 9.3 ± 1.1 | **6.3 ± 1.0** |
| Video | AIA (↑) | 80.6 ± 2.1 | 78.6 ± 3.9 | 80.4 ± 2.3 | 80.6 ± 2.5 | **90.1 ± 0.7** |
| | FAA (↑) | 69.8 ± 2.1 | 65.2 ± 3.0 | 72.0 ± 1.2 | 70.6 ± 0.6 | **82.9 ± 0.6** |
| | F (↓) | 11.5 ± 2.3 | 14.2 ± 1.3 | 8.9 ± 1.2 | 12.1 ± 1.8 | **7.7 ± 2.8** |
| Depth | AIA (↑) | 40.3 ± 3.8 | 44.5 ± 2.6 | 44.4 ± 4.8 | 46.2 ± 2.2 | **61.8 ± 2.7** |
| | FAA (↑) | 19.7 ± 4.4 | 23.5 ± 1.8 | 28.2 ± 4.8 | 29.1 ± 3.1 | **44.9 ± 1.0** |
| | F (↓) | 22.9 ± 1.4 | 24.5 ± 3.9 | 17.1 ± 1.1 | 18.1 ± 5.2 | **16.8 ± 3.8** |
| Audio | AIA (↑) | 46.7 ± 1.8 | 47.9 ± 4.5 | 46.4 ± 2.1 | 47.0 ± 0.9 | **65.7 ± 2.9** |
| | FAA (↑) | 19.5 ± 1.9 | 18.8 ± 2.9 | 20.5 ± 1.3 | 22.3 ± 3.0 | **47.9 ± 3.4** |
| | F (↓) | 28.9 ± 0.5 | 30.9 ± 6.1 | 27.5 ± 2.7 | 28.0 ± 2.1 | **18.9 ± 1.8** |
| Overall | AIA (↑) | 61.7 ± 2.4 | 61.9 ± 2.9 | 62.3 ± 2.9 | 63.7 ± 1.8 | **75.9 ± 1.6** |
| | FAA (↑) | 45.0 ± 2.7 | 43.8 ± 2.4 | 46.8 ± 2.5 | 48.3 ± 2.4 | **63.8 ± 1.3** |
| | F (↓) | 18.0 ± 1.1 | 19.7 ± 3.2 | 16.3 ± 1.4 | 16.8 ± 2.5 | **12.4 ± 1.7** |

Table 7: Results for three different class orders (mean±std).

AIA, FAA, and F, respectively. These results demonstrate the proposed COMM's robustness to class order variations.

**Impact of self-regularization.** The proposed knowledge aggregation across time within a modality consolidates old and new knowledge by self-regulating the previously aggregated prompts and the current ones. This approach eliminates the need to store previous individual prompts $\{P_m^1, \ldots, P_m^{t-1}\}$ and $\{Q_m^1, \ldots, Q_m^{t-1}\}$, which provides computational efficiency compared to other prompt-based continual learning methods Wang et al. (2022a); Zhou et al. (2022b); Smith et al. (2023a). This choice of implementation is particularly important in multimodal continual learning, where new tasks can introduce novel class subsets and novel modalities. As a result, more prompts and the associated trainable parameters are required to learn an expanding set of classes and modalities. This results in inaccurate knowledge retrieval (Table 3 in the main paper) and greater memory consumption (Figure 5 in the main paper).

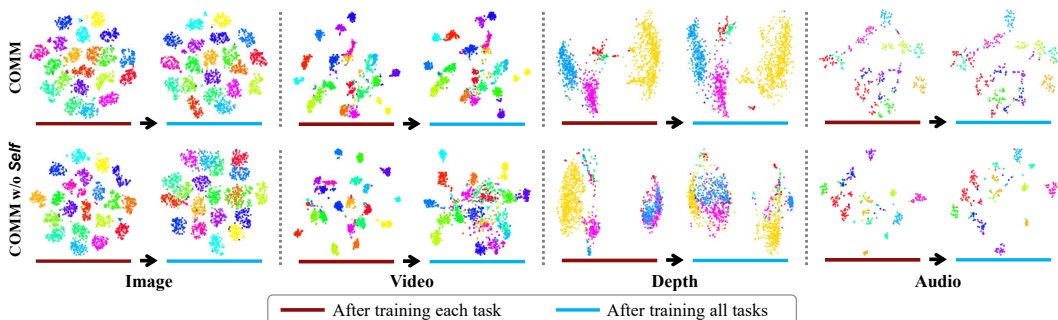

Figure 6: t-SNE visualization on the modality embeddings extracted from the proposed method, COMM, with and without including self-regularization (*Self*) of prompts. Each color corresponds to an individual class.

To clearly demonstrate the intrinsic maintainability of previous features by self-regularization (*Self*) after learning subsequent tasks, we visualize the embeddings obtained using the proposed method and the method without *Self* in Figure 6. We compared the embeddings of tasks extracted right after training each task with those extracted after training all subsequent tasks. The results show that the embeddings remain almost unchanged even after learning new tasks compared to the previously extracted embeddings, when applying the presented *Self*. However, after learning subsequent tasks, embeddings from the method without *Self* become mixed, especially in the video and depth modalities.

**Results on different numbers of learnable parameters.** We further analyze the scalability of the proposed method with respect to the number of learnable parameters. To adjust the number of learnable parameters, we increased the dimension of prompts and the layers of the pre-trained models for inserting prompts, following the methods in the previous studies Khattak et al. (2023);

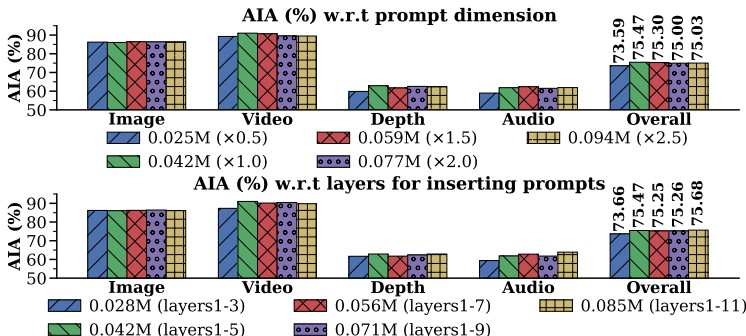

Figure 7: Results for the proposed method with varying learnable parameters by adjusting the prompt dimension (top) and the layers where prompts are inserted (bottom).

Zhou et al. (2022a). Figure 7 indicates that COMM maintains consistent overall AIA performance across different prompt dimensions and numbers of layers for prompt insertion.

**Results on unimodal continual learning.** While COMM is designed for multimodal continual learning, it can also be applied to unimodal learning scenarios. To validate this, we conducted additional continual learning experiments where each task consists of a single non-text modality (image or audio) paired with text. Specifically, we use ImageNet-100 for image-text tasks and ESC-50 for audio-text tasks, splitting each into five incremental tasks. As shown in Table 8, COMM significantly outperforms other prompt-based continual learning methods (L2P, S-liPrompts, AttriCLIP, and PGP) in both modalities, achieving higher average accuracy (AIA, FAA) and lower forgetting (F). This shows that the proposed method not only generalizes to multimodal settings but also improves learning stability in conventional unimodal tasks.

|  |  | L2P | S-liPrompts | AttriCLIP | PGP | COMM |
|---|---|---|---|---|---|---|
| Image | AIA (↑) | 76.9 | 78.4 | 77.3 | 79.5 | **85.8** |
|  | FAA (↑) | 68.6 | 67.5 | 65.7 | 70.2 | **79.3** |
|  | F (↓) | 10.3 | 13.7 | 14.6 | 11.7 | **8.1** |
| Audio | AIA (↑) | 44.5 | 41.3 | 41.7 | 44.8 | **61.7** |
|  | FAA (↑) | 25.0 | 19.3 | 23.5 | 26.2 | **49.0** |
|  | F (↓) | 24.3 | 27.5 | 22.8 | 23.3 | **15.9** |

Table 8: Results on unimodal continual learning