# OpenReview forum: "Mind the Interference: Towards Robust Continual Learning Across Modalities"
_ICLR.cc/2026/Conference — ICLR 2026 Conference Withdrawn Submission_

### Official Review · Reviewer_JndJ · 2025-10-24

**Soundness:** 2
**Presentation:** 1
**Contribution:** 2
**Rating:** 2
**Confidence:** 5

**Summary:**

This paper introduces a novel framework, COMM (COntinual Learning for Multiple Modalities), to address catastrophic forgetting when learning sequentially from diverse data types like images, video, audio, and text. Traditional continual learning methods, designed for a single modality, fail to handle the increased interference that arises when switching between different modalities.

To maintain previously acquired knowledge, the paper proposes a knowledge aggregation method that integrates prompts from relevant modalities with a self-regulating mechanism. In addition, the paper collects features mimicking old features to align the modality embedding.

**Strengths:**

1. The experimental settings (datasets, scenarios, etc.) are reasonable.

**Weaknesses:**

1. Motivation is Superficial and Unconvincing: The paper claims the problem is relevant for applications like autonomous driving and robotics (lines 62-64), but this justification is entirely hand-wavy. It fails to provide a single concrete, real-world scenario that maps to the proposed experimental setups (e.g., "random-modality continual learning"). Why would an autonomous vehicle learn disjoint sets of classes from images, then audio, then depth in a random sequence? The paper introduces a new benchmark but does not provide the necessary grounding to convince the reader of its practical value or relevance.
2. The technical novelty is minimal, as the method appears to be a repackaging of existing ideas. Aggregation is Not Novel: The core ideas of aggregating model components are well-established. Prompting Strategy is Derivative: The method relies on a modality-specific prompt pool. Given that prior work like L2P used a shared pool for unimodal tasks, dedicating separate prompt pools/histories for different modalities is a natural, almost trivial, design choice, not a novel insight.
3. The experiments are insufficient. The paper should compare with more SOTA methods. The latest baseline is from ICLR 2024, which was proposed in 2023.
4. The related works are somewhat old; the author should consider more recent works.
5. The writing is very poor and confusing. The paper uses many notations without giving a clear definition, especially in Section 3. This paper should be fully rewritten to meet the standards of a top conference. The authors do not need so many notations at all.
6. How do you use equation 3? The method is overly complex and unclear: The paper does not clearly describe the training process of R_t. Is it trained before the main task at each time step t, or is it trained jointly with the main task? This is a crucial implementation detail, but the paper is vague about it, making the method difficult to reproduce. This process of "training a tool first, and then using this tool to assist the main task" increases the complexity of the system and the training overhead.
7. The mechanism for generating features from past tasks (lines 245-247, 262-264) is a critical part of the method, yet it is described in a single, un-interrogated sentence referencing a paper. How are these means and covariances stored and updated? Per-class? Per-modality? How many samples are drawn? Is a single Gaussian a reasonable model for the feature distribution of an entire modality across multiple classes? This crucial assumption is neither justified nor ablated.
8. In Fig. 1 (b), why can the text modality not be a single task? Can the designed method solve this case?

**Questions:**

NA

---

### Official Review · Reviewer_RY7J · 2025-10-31

**Soundness:** 2
**Presentation:** 1
**Contribution:** 1
**Rating:** 2
**Confidence:** 4

**Summary:**

The paper investigates continual learning in multiple modalities, including image, video, audio, depth, and text. They propose a method, COMM (COntinual Learning for Multiple Modalities), for continuously learning tasks of different modalities, using a pre-trained vision-language model.

**Strengths:**

The paper proposes a new method in full modalities.

**Weaknesses:**

Do we really need a model to cover full modalities? And the Full modalities model starts from vision-language models? I really think continual learning of multiple modalities in a vision-language model is inappropriate.

The state ”Existing methods were designed to learn a single modality (e.g., image) over time” is inappropriate; the research in vision-language is hot, and the author should consider the related methods in vision-language continual learning.

Why select text as the center modality to align different modalities? Sound videos can also be taken as center modalities.
The presentation has weaknesses, with multiple redundancies and vagueness in the paper. For example, I get no information about the proposed methods in the abstract.

The baseline is outdated, and there is a lack of sufficient surveys for multi-modal continual learning.

**Questions:**

See Weakness.

---

### Official Review · Reviewer_cMuT · 2025-10-31

**Soundness:** 2
**Presentation:** 3
**Contribution:** 2
**Rating:** 4
**Confidence:** 4

**Summary:**

This submission introduces COMM (Continual Learning for Multiple Modalities), a framework designed to prevent catastrophic forgetting in continual learning settings involving sequentially presented data from various modalities (image, video, audio, depth, and text).

The method adapts a pre-trained vision-language model (like CLIP) using learnable prompts. COMM's key innovation is a two-part aggregation strategy for these prompts:

(i) Intra-modal Self-Regularization: It stabilizes knowledge acquisition for a single modality over sequential tasks.

(ii) Inter-modal Relevance-Guided Blending: It is the most interesting and central contribution, which uses a learned relevance function (a classifier) to generate weights for blending prompts from previously seen modalities, effectively reducing interference between different modalities.

**Strengths:**

The primary strength is the modality-aware prompt blending, which explicitly handles inter-modal interference—a unique challenge in this new multi-modal continual learning setting. COMM achieves significantly superior performance in both modality-specific and modality-agnostic continual learning scenarios compared to existing methods.

**Weaknesses:**

1/ In the literature review, the authors should talk about multi-modality (or multi-task) continual learning.

2/ The main weakness lies in the evaluation design. As COMM is claimed to be the first method of its kind, the comparison baselines are all adapted uni-modal continual learning methods. The evaluation would be stronger with comparisons against adapted multi-task continual learning or more general multi-modal frameworks.

3/ The modality-agnostic approach scenario may not be applicable: the pre-processing pipelines for different data modalities are very different, so the modality information is almost always available.

**Questions:**

For the modality-aware setting, how do the authors inject modality information into the baseline methods?

---

### Note · Authors · 2025-11-26

I have read and agree with the venue's withdrawal policy on behalf of myself and my co-authors.